# Survey on the Indoor Thermal Environment and Passive Design of Rural Residential Houses in the HSCW Zone of China

**Jingwen Rui** [1,2], **Huibo Zhang** [1,2,*], **Chengnan Shi** [2], **Deng Pan** [3,4], **Ya Chen** [1,2] and **Chunyu Du** [1]

1   Department of Architecture, School of Design, Shanghai Jiao Tong University, Shanghai 200240, China; ruijingwen@sjtu.edu.cn (J.R.); madilyn@sjtu.edu.cn (Y.C.); cydu@sjtu.edu.cn (C.D.)
2   China-UK Low Carbon College, Shanghai Jiao Tong University, Shanghai 200240, China; chengnanshi@sjtu.edu.cn
3   Materials Genome Institute, Shanghai University, Shanghai 200444, China; DPan_MGI@shu.edu.cn
4   Center for Advanced Metallic Materials, Yangtze Delta Region Institute of Tsinghua University, Jiaxing 314006, China
*   Correspondence: zhanghuibo@sjtu.edu.cn

**Abstract:** Despite their high energy consumption, rural residential houses in the hot summer and cold winter (HSCW) zone still have a generally poor indoor thermal environment. The objective of this study was to understand the current status of the indoor thermal environment for rural residential houses in the HSCW zone and analyze its cause in order to develop some strategies for improvement through passive design of the building envelope. Face-to-face questionnaires and interviews, air-tightness testing, and temperature and humidity monitoring were conducted to understand the building envelope, energy consumption, and indoor thermal environment. Then, some passive design strategies were simulated, including the application of functional interior materials such as hygroscopic and phase change materials. An overall passive design for the building envelope can increase the room temperature by 3.6 °C, reduce the indoor relative humidity by 12% in the winter, and reduce the room temperature by 4.4 °C in the summer. In addition, the annual energy-saving rate can reach ~35%.

**Keywords:** rural residential houses; hot summer and cold winter (HSCW) zone; indoor thermal environment; passive design; phase change material; energy consumption

## 1. Introduction

In China, the hot summer and cold winter (HSCW) zone mainly includes the middle and lower reaches of the Yangtze River and its surrounding areas. In this zone, the average temperature in the hottest month is 25–30 °C; it is the hottest area in the world at this latitude except for arid deserts. The average temperature in the coldest month is 2–8 °C, which is the coldest in the world at this latitude [1]. This zone is also very humid; the relative humidity in most cities is 75%–80% and sometimes even 95%–100% [2]. These climatic features make the indoor thermal environment improvement and passive design of rural residential houses in the HSCW zone more difficult.

In rural China, the improving economic level and increases in the number and frequency of electrical appliances caused rural residential commercial energy consumption to reach 243 million tons of coal equivalent (TCE) in 2017; this accounted for 25.31% of the total commercial energy consumption of buildings in China [3]. Based on the per capita energy intensity of rural houses, the energy consumption per household in the southern region, including the HSCW zone, increased 49.4% from 0.89 TCE in 2006 to 1.33 TCE in 2014 [4]. The proportion of rural commercial energy consumption

in cities of southern China such as Shanghai, Zhejiang Province, Jiangsu Province, and Hunan Province has exceeded 70% [4]. In the southern region, the total heating energy consumption of solid fuels is about 29 million TCE, which is lower than that in the northern region. The total cooling electricity consumption of the southern region is about 30 billion kWh, which accounts for 80% of the national cooling electricity consumption [4]. In general, rural residential housing in the HSCW zone has great potential for energy use, which has deterred the achievement of targets for sustainable development. Despite the high energy consumption, rural residential houses in the HSCW zone still have a generally poor indoor thermal environment. The average indoor temperature of rural residential houses in the HSCW zone can reach 29.7 °C in the summer, which indicates that the indoor thermal environment falls beyond the 80% acceptability range of the adaptive model defined by ASHRAE standard 55 [5]. A neutral temperature of 10.7 °C and 80% acceptable lower temperature of 4.7 °C were obtained for bedrooms in rural areas of China's HSCW zone, which further demonstrates the poor indoor thermal environment [6]. In Shanghai, Changsha, and Chongqing, houses were observed to have a high relative humidity of 60%–100% [7].

Many researchers have focused on the passive design of building envelopes in the HSCW zone [8–13]. Yang and Peng [14] renovated an existing farmhouse in Chongqing by insulating the exterior walls and roof; after the renovation, the average indoor temperature increased by 2 °C in the winter and decreased by 1.2 °C in the summer. Fu et al. [15] found that decreasing the air changes per hour (ACH) can reduce the annual heating and cooling load and had a remarkable effect on the heating load. Zhou et al. [16] studied the influence of a building's air tightness level on the annual dynamic energy consumption of buildings and found that reducing the ACH from 1 to 0.1 could reduce the annual electricity consumption by 15%. Costanzo et al. [17] found a tremendous reduction in the natural ventilation potential—expressed in terms of window opening hours—with outdoor pollution threshold values. Chow et al. [18] simulated a building in the HSCW zone and introduced a series of measures such as double-glazed windows and reducing the heat transfer coefficients of the exterior walls and roof; they found that these measures reduced the total primary energy by 40%. Yao et al. [19] discovered that, by applying proper passive design measures in the HSCW zone, the non-heating and -cooling periods can be extended.

However, very few studies have focused on rural residential houses in the HSCW zone. Most of the existing research only considered a single part of the building envelope, and few studies considered the overall passive design of the building envelope. In addition to addressing the indoor temperature and energy conservation in the HSCW zone, the problem of high humidity also needs to be considered. Phase change materials (PCMs) are a new type of energy storage material that can reduce the peak load and energy consumption [20], and hygroscopic materials are a functional material that can help suppress peaks and mitigate fluctuations in the indoor humidity with its porous structure [21]. Few studies have considered applying these two functional materials to passively improving the indoor thermal environment of rural residential houses in the HSCW zone.

The objective of this study was to understand the current status of the indoor thermal environment for rural residential houses in the HSCW zone and analyze its cause in order to develop some strategies for improvement through passive design of the building envelope.

## 2. Methods

### 2.1. Face-To-Face Questionnaire and Interview

Face-to-face questionnaires and interviews were conducted in 2016. The building envelope and energy consumption of 80 rural residential houses in Anji County, Zhejiang Province, were surveyed. The contents of the questionnaire mainly included the building characteristics, housing equipment and their use, and the lifestyle, energy use, and income of the residents.

## 2.2. Field Measurement

### 2.2.1. Air-Tightness Testing

The overall air-tightness performance of a building was measured with an OM Solar air-tightness tester (Figure 1a) by the decompression method. The measurement instrument was used to directly read the average flow under positive pressure and negative pressure at a pressure difference of 50 Pa. The ACH at a pressure difference of 50 Pa was converted to the natural condition as per DB11/T 555-2015 [22].

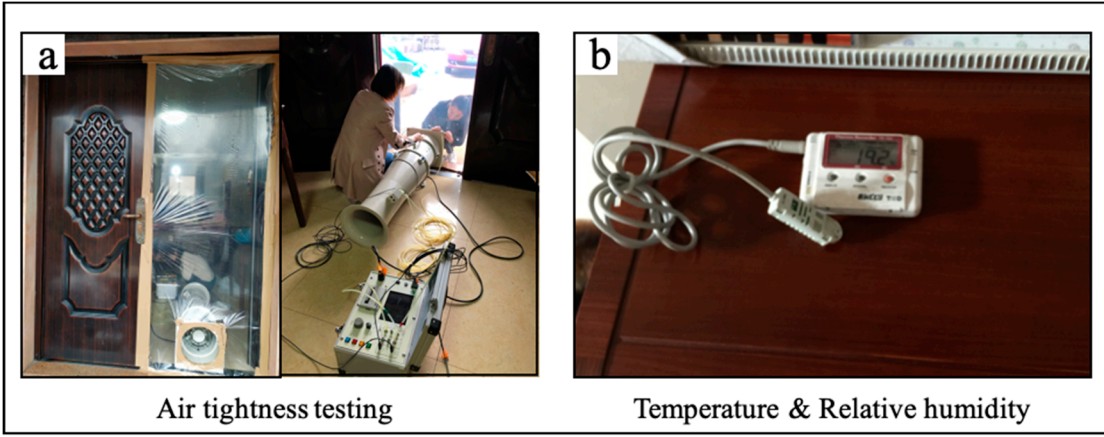

**Figure 1.** Field measurements: (**a**) air-tightness testing and (**b**) temperature and relative humidity measurement.

### 2.2.2. Temperature and Humidity Monitoring

From April 2016 to March 2017, the indoor and outdoor air temperature and the humidity were continuously measured for two households with similar building structures and living habits in Dazhuyuan Village, Anji County, Zhejiang Province with the TR-76Ui instrument (T&D Corp.), which has a measurement accuracy of ±0.5 °C, ±5% relative humidity, and ±50 ppm and valid range of 0–55 °C, 10%–95% relative humidity, and 0–9999 ppm (Figure 1b). These two houses had different types of window glass: one used single-glazed windows, and the other used double-glazed windows. For convenience, the single-glazed and double-glazed households were designated as Houses A and B, respectively.

### 2.2.3. Fungal Index

The fungal index, which was developed by Abe after a large number of experiments [23–25], was used to evaluate the possibility of fungal contamination of rural residential houses in the HSCW zone. The fungal index is a biological climate parameter that expresses the fungal response unit (ru/week) for measuring the fungal growth response during the exposure period (week) at a certain temperature and relative humidity. The temperature affects the fungal index. At 0–28 °C, the fungal index increases with the temperature. Above 28 °C, the fungal index decreases with an increasing temperature. The relative humidity has a great influence on the fungal index. The fungal index increases with the relative humidity. When the relative humidity is below 70%, almost no fungal contamination is generated. Usually, a fungal index of 3 ru/week or more indicates the possibility of fungal contamination. A fungal index of 10 ru/week or more indicates a high possibility of fungal contamination.

### 2.3. Simulation

#### 2.3.1. Base Model

In this study, a base model was built for the prototype of a three-story house in Anji County with the DesignBuilder software. Figure 2 shows the visualization and layout of the base model. The indoor thermal environment and energy consumption of the base model were simulated with the EnergyPlus software. The conduction transfer function (CTF) algorithm was used for calculation. The specific construction of the building's envelope was set according to the results of questionnaires and on-site inspection. The properties of each material were obtained from EnergyPlus software. Table 1 presents the thermal properties of the base model's envelope. The window-to-wall ratios (WWRs) in the southern and northern directions were 15.87% and 12.93%, respectively. There were no external windows in the eastern and western directions. In addition, ACH was set to 1.5 h$^{-1}$ based on the results of the air-tightness test. Based on the results of the questionnaire surveys, the heating and cooling periods were set to be 8:00 a.m. – 12:00 p.m. for November 21–February 10 and 5:00 p.m. –8:00 a.m. for June 11–September 30, respectively. The heating and cooling temperatures were set to 22 °C and 26 °C, respectively. In addition, in order to better verify the simulation results and the measured results, the outdoor temperature and humidity data used in the simulation of this study are derived from the measured results.

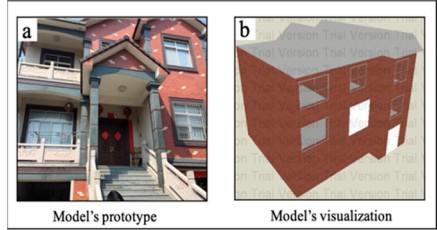 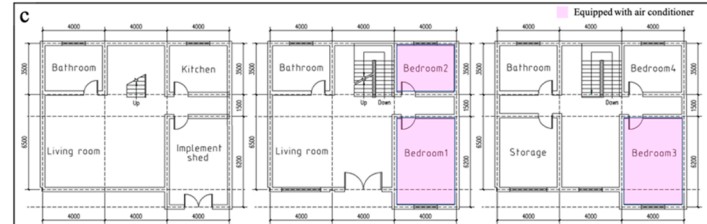

**Figure 2.** Model's (**a**) prototype, (**b**) visualization, and (**c**) layout in Anji County.

**Table 1.** Thermal properties of the base model's envelope.

| System | Construction (From Outside to Inside) | Thickness [mm] | Thermal Conductivity [W/m·K] | Specific Heat [J/kg·K] | Density [kg/m³] |
|---|---|---|---|---|---|
| Exterior wall | Ceramic | 10 | 1.40 | 840 | 2500 |
| | Brick | 240 | 0.72 | 840 | 1920 |
| | Mortar | 20 | 0.88 | 896 | 2800 |
| | Total | 260 | Heat transfer coefficient: 1.875 W/m²·K | | |
| Interior wall | Mortar | 10 | 0.88 | 896 | 2800 |
| | Brick | 240 | 0.72 | 840 | 1920 |
| | Mortar | 10 | 0.88 | 896 | 2800 |
| | Total | 260 | Heat transfer coefficient: 1.623 W/m²·K | | |
| Window | Single glazing | 3 | Heat transfer coefficient: 5.894 W/m²·K | | |
| Roof | Sloping roof with wooden structure (no insulation) | | Heat transfer coefficient: 1.565 W/m²·K | | |

#### 2.3.2. Composite Material: CMPCM-15

A composite material was used to passively improve the indoor thermal environment and reduce energy consumption. Gypsum board was used as a hygroscopic material for interior finishing and was mixed with 15 wt % microencapsulated PCM (phase change temperature: 26 °C, enthalpy: 120 kJ/kg) to prepare composite microencapsulated PCM (CMPCM-15). The conduction finite difference (ConFD) algorithm was used to simulate the PCM [26]. All properties of CMPCM-15 were tested experimentally [27]. The density of CMPCM-15 is 1076 kg/m³, of which its thermal conductivity and



specific heat are 0.49 W/m·K and 1317 J/kg·K, respectively. For moisture properties, the vapor resistance factor and effective moisture penetration of CMPCM-15 are, respectively, 8.594 and 7.1 mm.

The moisture transfer in CMPCM-15 was calculated with the effective moisture penetration depth (EMPD) algorithm, which is a simplified lumped moisture model that simulates moisture storage and release from interior surfaces [28]. The equilibrium moisture sorption isotherm can be defined by the following general equation:

$$U = a\varphi^b + c\varphi^d \tag{1}$$

where $U$ [kg/kg] is the moisture content defined as the mass fraction of water contained in a material, $\varphi$ [0,1] is the surface air relative humidity, and $a$, $b$, $c$, $d$ are the coefficients that define the relationship between the material's moisture content and the surface air relative humidity. In this study, the moisture content of CMPCM-15 was tested experimentally. The average moisture content during the sorption and desorption processes was used for fitting. The fitting curve is shown in Figure 3a.

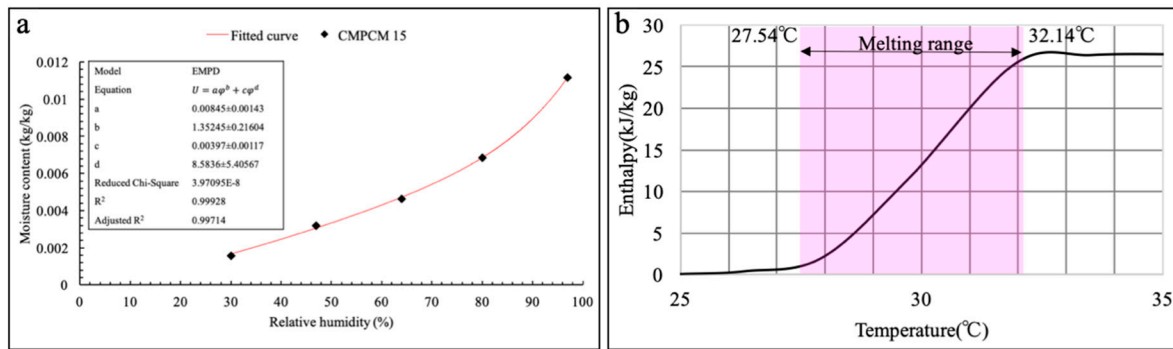

**Figure 3.** (**a**) Isothermal equilibrium moisture content curve and (**b**) Temperature–enthalpy curve of CMPCM-15.

The thermal properties of CMPCM-15 were analyzed with a differential scan calorimeter (DSC). A sample was tested at a rate of 5 °C/min in the temperature range of −20 °C to 80 °C. The melting temperature range was about [27.54 °C, 32.14 °C], and the latent heat of CMPCM-15 was about 26.5 kJ/kg. The temperature–enthalpy curve is shown in Figure 3b.

### 2.3.3. Case Details

Building upon the base model, several passive design strategies were proposed as per JGJ134-2010 [29] to improve the indoor thermal environment and energy conservation, as presented in Table 2. The cases had six parts: the exterior walls, interior walls, windows, roof, air tightness, and overall design of building envelope. For the exterior walls, external and internal thermal insulation systems were considered, and expanded polystyrene (EPS) was used as the insulation material. With the exception of the different thicknesses of the CMPCM-15 layer (10 mm and 20 mm) on the inner surface of the exterior wall, the 20 mm CMPCM-15 layer on the inner surface of the interior wall was also under consideration, and both were combined with the insulation system. The position and the thickness of the PCMs must be carefully selected in order to ensure their effective operation [30]. For the other parts, double-glazed windows and roof insulation were all considered. In this study, the improvement of air tightness was applied in all rooms of the model house. On the basis of better air-tightness performance, natural ventilation was also considered; however, it was only considered for the rooms that did not use the heating and cooling equipment. In addition to the natural ventilation during the days in the spring and autumn, times during the afternoon in the winter and nights in the summer were also taken into account. For the overall design of the building envelope, the individual passive design strategies described above were combined.

**Table 2.** Parameters of the envelope construction for the considered cases.

| Case | Code | System | Construction (From Outside to Inside) | Heat Transfer Coefficient [W/m²·K] | |
|---|---|---|---|---|---|
| | | | | Model Value | JGJ134-2010 Limit |
| 1 | Ex | Exterior wall | 10 mm ceramic, 30 mm EPS, 240 mm brick, 20 mm mortar | 0.707 | 0.8 |
| 2 | In | Exterior wall | 10 mm ceramic, 240 mm brick, 30 mm EPS, 20 mm mortar | 0.707 | 0.8 |
| 3 | 10 mm CMPCM-15 | Exterior wall | 10 mm ceramic, 240 mm brick, 20 mm mortar, 10 mm CMPCM-15 | 1.806 | 0.8 |
| 4 | 20 mm CMPCM-15 | Exterior wall | 10 mm ceramic, 240 mm brick, 20 mm mortar, 10 mm CMPCM-15 | 1.742 | 0.8 |
| 5 | All (10 mm CMPCM-15) | Exterior wall | 10 mm ceramic, 240 mm brick, 20 mm mortar, 10 mm CMPCM-15 | 1.806 | 0.8 |
| | | Interior wall | 10 mm mortar, 240 mm brick, 10 mm mortar, 10 mm CMPCM-15 | 1.317 | |
| 6 | All (20 mm CMPCM-15) | Exterior wall | 10 mm ceramic, 240 mm brick, 20 mm mortar, 20 mm CMPCM-15 | 1.742 | 0.8 |
| | | Interior wall | 10 mm mortar, 240 mm brick, 10 mm mortar,20 mm CMPCM-15 | 1.522 | |
| 7 | Ex + All (20 mm CMPCM-15) | Exterior wall | 10 mm ceramic, 30 mm EPS, 240 mm brick, 20 mm mortar, 20 mm CMPCM-15 | 0.699 | 0.8 |
| | | Interior wall | 10 mm mortar, 240 mm brick, 10 mm mortar, 20 mm CMPCM-15 | 1.522 | |
| 8 | In + All (20 mm CMPCM-15) | Exterior wall | 10 mm ceramic, 240 mm brick, 30 mm EPS, 20 mm mortar, 20 mm CMPCM-15 | 0.699 | 0.8 |
| | | Interior wall | 10 mm mortar, 240 mm brick, 10 mm mortar, 20 mm CMPCM-15 | 1.522 | |
| 9 | Double layer window | Window | Double low-E 6 mm/13 mm air | 1.761 | 4.0 |
| 10 | Roof insulation | Roof | Wooden structure pitched roof with 60 mm EPS | 0.425 | 0.5 |
| 11 | ACH = 1 + NV | Air tightness | From 11/1 to 2/28, 11:00 a.m. – 1:00 p.m., ACH = 5 h⁻¹; From 6/1 to 9/30, 7:00 p.m.– 7:00 a.m., ACH = 5 h⁻¹; From 3/1 to 5/30 and from 10/1 to 10/30, 8:00 a.m. – 5:00 p.m., ACH = 5 h⁻¹; Other times, ACH = 1 h⁻¹; | | 1.0 h⁻¹ |
| 12 | Combined (Ex) | | Case 7 + Case 9 + Case 10 + Case 11 | | |
| 13 | Combined (In) | | Case 8 + Case 9 + Case 10 + Case 11 | | |

The indoor thermal environment of each case was analyzed according to the annual daily room temperature, annual daily indoor relative humidity, hourly indoor temperature, and relative humidity for the coldest and hottest weeks. The energy conservation was analyzed according to the energy-saving rate:

$$\Phi = \left(1 - \frac{E_{bld,des}}{E_{bld,ref}}\right) \times 100\% \tag{2}$$

where $\Phi$ is the energy-saving rate, $E_{bld,\,des}$ [kWh] is the comprehensive energy consumption for annual heating and cooling of the design building, and $E_{bld,\,ref}$ [kWh] is the comprehensive energy consumption for annual heating and cooling of the reference building. In this study, $E_{bld,\,des}$ and $E_{bld,\,ref}$ were set to the comprehensive energy consumption for annual heating and cooling of the base case and the case being considered, respectively.

## 3. Results and Discussion

### 3.1. Face-To-Face Questionnaire and Interview

#### 3.1.1. Building Characteristics

About 93% of the 80 households surveyed lived in houses that were two or more stories and were built before 2010. Of these houses, 89% had residential areas greater than 200 m². The average residential area was 273 m², but the houses had many vacant rooms. Some houses had a shape factor of 0.5. According to JGJ134-2010 [29], the shape factor limit is 0.55 for houses with three stories or less. The exterior walls mostly comprised 240 mm thick brick as the load-bearing material, and up to 90% of the exterior walls had no insulation. For the windows, 96% of the houses used single-glazed 3 mm thick glass windows. The roofs were very simple with a pitched wooden structure and no insulation.

These results show that the vast majority of rural houses in the HSCW zone have thermal properties that do not satisfy JGJ134-2010 [29]. The poor thermal performance of the building envelope caused the high energy consumption and poor indoor thermal environment of the rural houses.

### 3.1.2. Heating and Cooling Equipment and Its Energy Usage

Up to 89% of the houses were equipped with air conditioners for heating and cooling. More than 85% of the households chose to use air conditioning for cooling from late June to early September, and more than 70% of the households used it for heating from late November to early January, as shown in Figure 4. The demand for cooling in the summer was much higher than the demand for heating. The average household electricity fee was 139.2 Chinese yuan (CNY) per month, and the average household gas fee was 75 CNY/month. According to the China Family Panel Studies, if an average of roughly four residents per rural household is assumed, a rural household in China would require 90 kWh/month to meet basic needs [31]. Thus, the monthly electricity fee for a rural household in China would be 55 CNY/month. Therefore, the energy consumption fee is relatively high for rural houses in the HSCW zone.

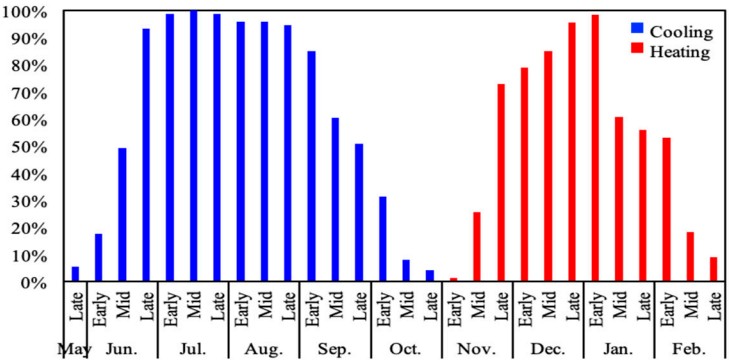

**Figure 4.** Air conditioner usage schedule.

In this study, July and August were used to represent the cooling season. December and January were used to represent the heating season. The average electricity fee for a household in the cooling season was 188.7 CNY/month, which is much higher than that in the heating season. In addition, the energy consumption cost during the cooling season varied greatly in the area, which was due to the difference in living habits with regard to air conditioner use. The average electricity fee for a household during the heating season was only 123.5 CNY/month. As rural households pursue a more comfortable indoor thermal environment, the heating energy consumption in this zone may increase in the future.

### 3.2. Field Measurement

### 3.2.1. Air-Tightness Testing

The ACH was about 1.42 h$^{-1}$, which is much higher than the limit set by JGJ134-2010 (1.0 h$^{-1}$) [29]. An onsite investigation showed that wooden roof gaps, window frame gaps, and air conditioning holes are very common, as shown in Figure 5. Thus, the houses had a poor air-tightness performance. The shoddy quality was mainly because the rural houses were constructed by the residents themselves. Indoor and outdoor air can directly exchange through the building gaps. Thus, reducing air infiltration can reduce the energy consumption needed for indoor temperature control [32].

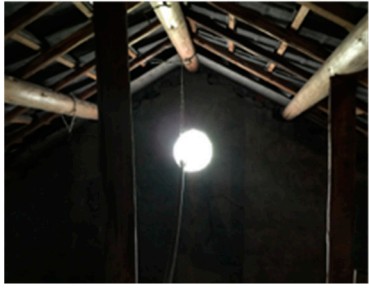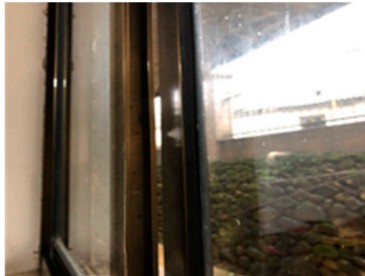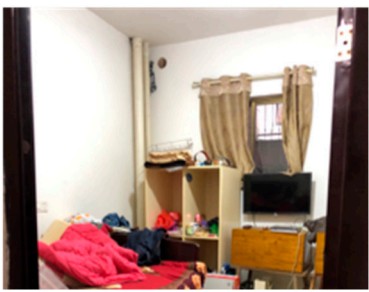

**Figure 5.** Roof, window, and wall gaps (from left to right).

### 3.2.2. Temperature and Humidity Monitoring

Figure 6 shows the temperature and humidity monitoring results of two households for almost 1 year. The indoor vertical temperature difference of House A in the summer reached 15 °C, which is much greater than that of House B. In the hottest month, the 2F bedroom of House A had a lower room temperature and relative humidity than the 2F bedroom of House B. This may be because House A used the air conditioner for a longer time than House B. During other times in the summer when the air conditioner was not used, House B had a lower average indoor temperature for this room than House A. In the coldest month, the average indoor temperature of the same room in House B was 0.3–2.3 °C higher than that of House A. Thus, double-glazed windows seem to improve the indoor thermal comfort of the house to some extent.

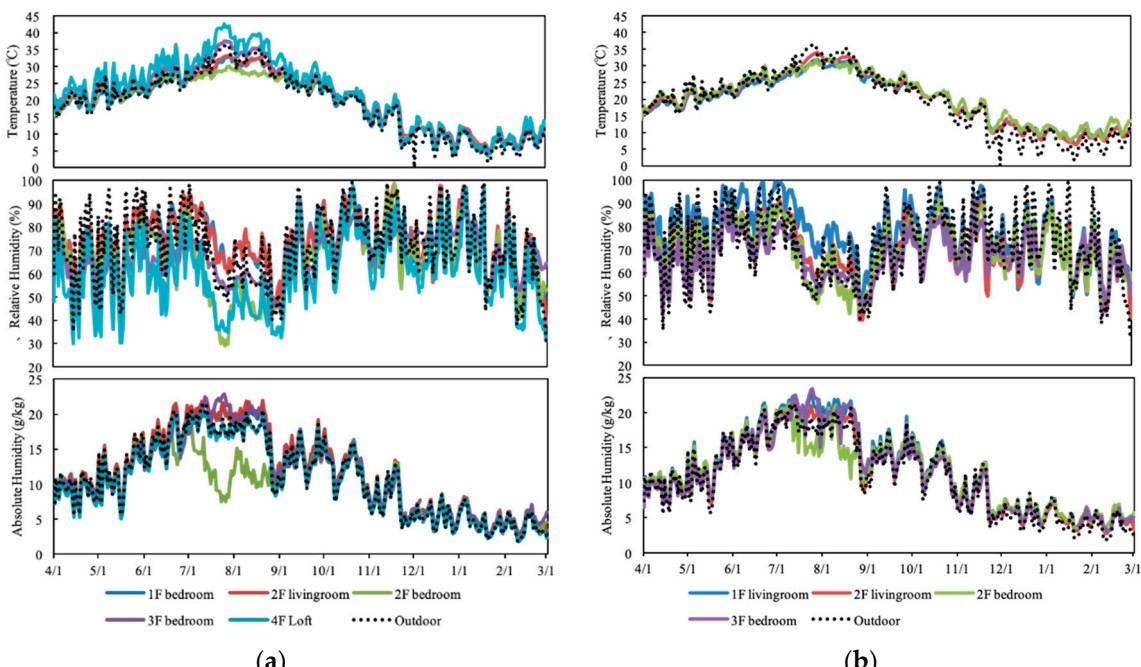

**Figure 6.** Daily average temperature and humidity variations of (**a**) House A, (**b**) House B.

### 3.2.3. Fungal Growth

Figure 7 shows the calculated fungal index of the two houses. The fungal index was always greater than 10 ru/week for both houses, which indicates a high possibility of fungal contamination. Table 3 presents the percentage of the whole year when the fungal index was 10 ru/week or more and the relative humidity was 70% or more. All rooms of House B had a higher possibility of fungal contamination than House A, except for the 2F living room. Most rooms of House B had a higher fungal index than the rooms of House A, especially in the summer. This is because House A used the

air conditioner longer than House B in the summer. Thus, House A generally had a lower relative humidity than House B. During other times, House B and House A had very similar values for the fungal index. Figure 8 shows some of the moisture damage, which was very common in the rural houses of this area. The results of the interviews validated the calculated results with the fungal index and indicated very serious fungal contamination of rural houses in this area.

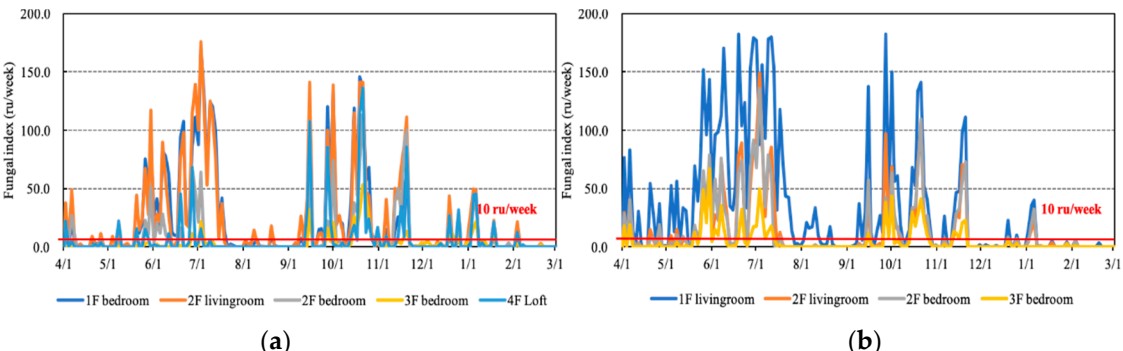

| (a) | (b) |

**Figure 7.** Yearly fungal index of (**a**) House A, (**b**) House B.

**Table 3.** Percentage of the whole year with a fungal index of 10 ru/week or greater and relative humidity values of 70% or greater.

| House A | 1F Bedroom | 2F Living Room | 2F Bedroom | 3F Bedroom | 4F Loft |
|---|---|---|---|---|---|
| Relative humidity ≥ 70% | 59.4% | 60% | 42.7% | 41.2% | 31.9% |
| Fungal index ≥ 10 ru/week | 35.9% | 37.7% | 19.8% | 9.6% | 15.6% |
| **House B** | **1F Living Room** | **2F Living Room** | **2F Bedroom** | **3F Bedroom** | |
| Relative humidity ≥ 70% | 68.7% | 49.6% | 51.9% | 41.5% | |
| Fungal index ≥ 10 ru/week | 50.3% | 27% | 27% | 18% | |

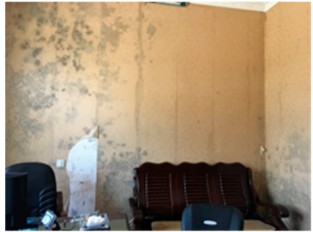 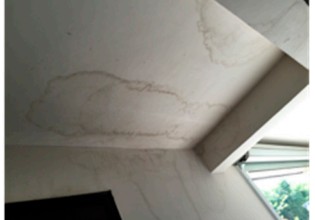 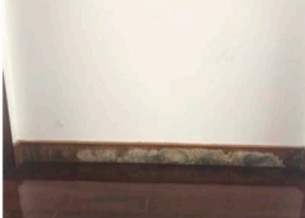

**Figure 8.** Areas with moisture damage.

*3.3. Simulation*

3.3.1. Indoor Thermal Environment

In order to analyze the influence of different building envelope constructions on the indoor thermal environment, a bedroom with no air conditioner on the third floor was considered as the analysis object, and the current situation for the envelope structure of rural residential houses in the HSCW zone was taken as the base case. The changes in the indoor temperature and relative humidity of various building envelope constructions relative to the base case and without the use of heating or cooling equipment were considered.

Daily Room Temperature and Average Indoor Relative Humidity for the Whole Year

Figure 9 shows the daily average indoor and outdoor temperatures and relative humidity over the whole year. It can be seen that there is substantial agreement between the simulation and measurement

results. In the base case, the room temperature fluctuated around 8 °C in the winter and was only about 1 °C higher than the outdoor temperature. In the summer, the room temperature fluctuated around 31 °C and was about 1 °C higher than the outdoor temperature. During the transition seasons, the room temperature fluctuated around 20 °C and was close to the outdoor temperature. The indoor relative humidity fluctuated more in the winter than in the summer. High humidity (relative humidity ≥ 70%) occurred with a frequency of 39%. Figure 9 clearly shows the improvement in the indoor thermal environment after the overall design of the building envelope.

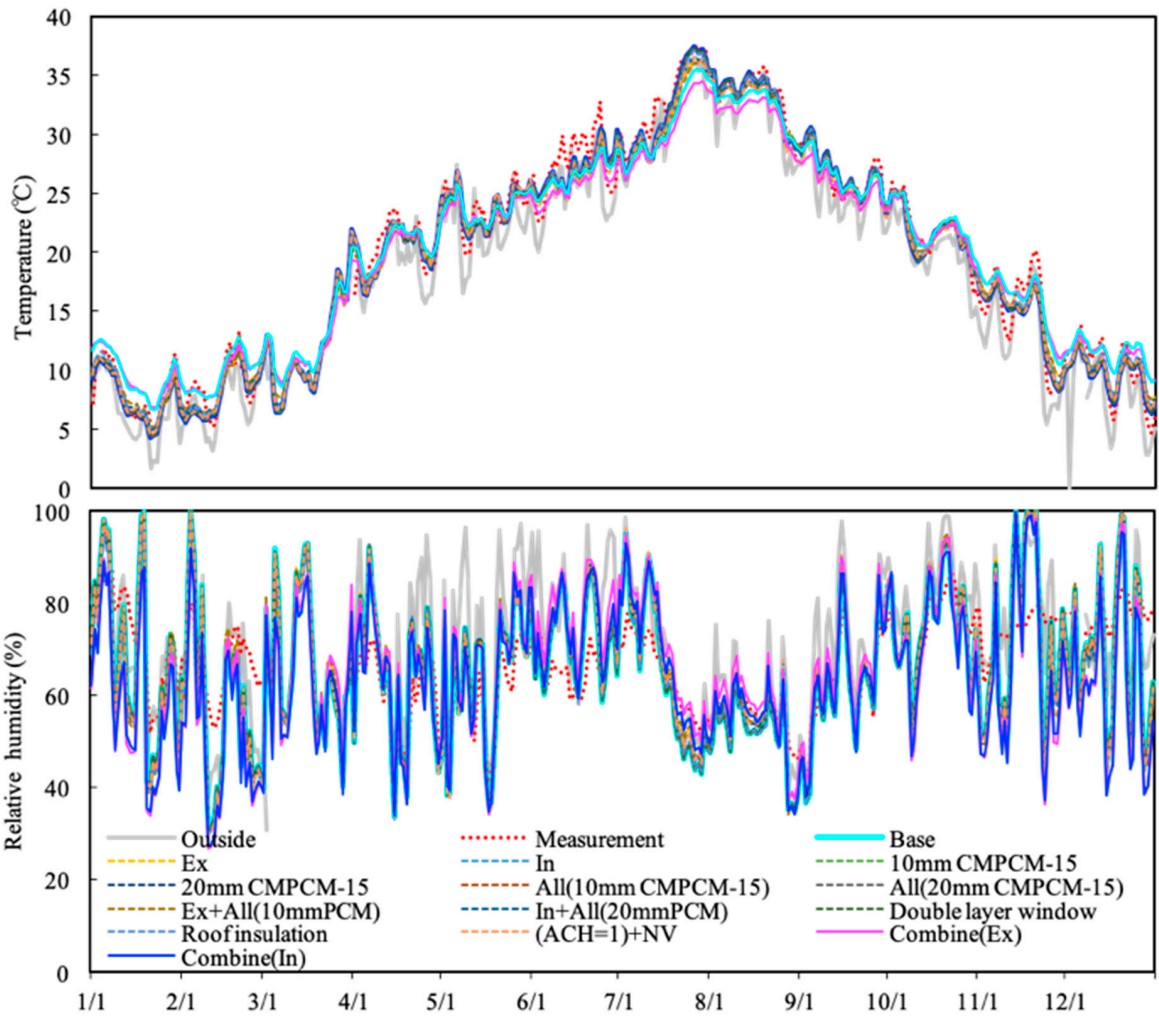

**Figure 9.** Daily room temperature and relative humidity for each case over the whole year.

Table 4 presents the statistical values (average and standard deviation) of the room temperature ($T_r$) for each case in the winter (December–February) and summer (June–August). In the winter, Case Ex and Case In greatly increased the room temperature to $T_r$ values of 8.9 ± 1.8 °C and 8.9 ± 2.0 °C, respectively. For Case Roof Insulation, $T_r$ reached 9.0 ± 2.1 °C. Thus, an insulation system is crucial to increasing the room temperature. Whether the 10 mm CMPCM-15 or 20 mm CMPCM-15 were added into the exterior walls and interior walls, $T_r$ was increased by 0.1 °C compared with the base case in the winter. For the Case Combined (Ex) and Case Combined (In), $T_r$ in the winter reached 10.1 ± 1.7 °C and 10.3 ± 1.8 °C, respectively. In the summer, it is worth noting that Case (ACH = 1) + NV provided significant temperature regulation effects, and its $T_r$ was only 30.1 ± 3.6 °C. For Case Combined (Ex) and Case Combined (In), $T_r$ reached 29.1 ± 3.2 °C and 30.0 ± 3.3 °C, respectively. Conspicuously, no matter in the winter or summer, the overall passive design increased the room temperature the most.

Table 5 presents the statistical values (average and standard deviation) of the indoor relative humidity ($RH_{in}$) for each case in the winter (December–February) and summer (June–August). In the winter, Case 10 mm CMPCM-15 and Case 20 mm CMPCM-15 reduced the average $RH_{in}$ by 0.1%, while Case All (10 mm CMPCM-15) and Case All (20 mm CMPCM-15) reduced the average $RH_{in}$ by 0.2%. It can be concluded that adding the position of the CMPCM-15 plays a role in humidity reduction. For Case Combined (Ex) and Case Combined (In), the $RH_{in}$ reached 56.2 ± 16.8% and 57.6 ± 16.7%, respectively, in the winter. Obviously, the overall passive design was best able to reduce $RH_{in}$. In the summer, in addition to Case Roof insulation, the other cases all increased the average $RH_{in}$ or remained stable. This is because, if the outdoor absolute humidity is the same, lowering the indoor temperature increases the average $RH_{in}$; however, most cases had reduced fluctuations in $RH_{in}$.

Daily Room Temperature and Average Indoor Relative Humidity During the Coldest and Hottest Weeks

In order to analyze the impact of each case on the indoor thermal environment in the winter and summer in more detail, the hourly indoor temperature and relative humidity were also considered for the coldest week (January 20–January 26) and hottest week (July 23–July 29).

Case Ex and Case In

As shown in Figure 10, compared to the base case, Case Ex and Case In increased the lowest room temperature ($T_l$) in the coldest week by 0.9 °C and 0.2 °C, and increased $T_r$ by up to 1.2 °C and 1.5 °C, respectively. Besides, Case Ex and Case In reduced $RH_{in}$ by 3.7% and 3.4%, respectively. In the hottest week, Case Ex and Case In reduced the peak room temperature ($T_p$) by 1 °C and 0.1 °C and reduced $T_r$ by up to 1.8 °C and 1.5 °C, respectively. Regardless of the season, Case Ex was better able to regulate the temperature and humidity than Case In. This is because the external thermal insulation system has the main body structure of the exterior wall inside the insulation layer, so more heat and humidity can be stored and released.

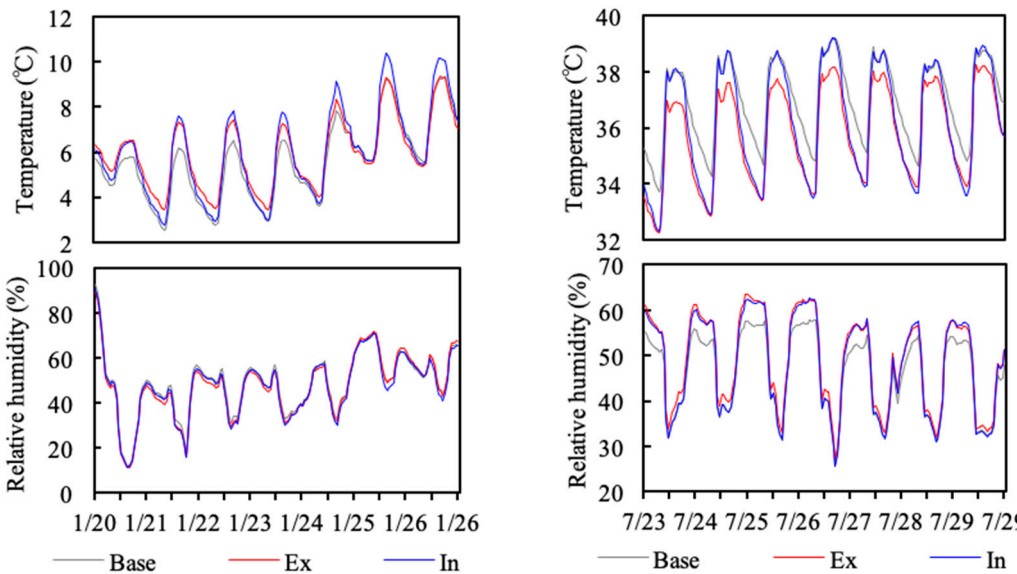

**Figure 10.** Hourly indoor temperature and relative humidity of Case Ex and Case In during the coldest week (left) and hottest week (right).

**Table 4.** Average (AVE) and standard deviation (STD) of the monthly room temperature in the winter and summer.

| | | Base | | Ex | | In | | 10 mm CMPCM-15 | | 20 mm CMPCM-15 | | All (10 mm CMPCM-15) | | All (20 mm CMPCM-15) | | Ex + All (20 mm CMPCM-15) | | In + All (20 mm CMPCM-15) | | Double-Layer Window | | Roof Insulation | | (ACH = 1) + NV | | Combined (Ex) | | Combined (In) | |
|---|---|---|---|---|---|---|---|---|---|---|---|---|---|---|---|---|---|---|---|---|---|---|---|---|---|---|---|---|---|
| | | AVE. | STD. | AVE. | STD. | AVE. | STD. | AVE. | STD. | AVE. | STD. | AVE. | STD. | AVE. | STD. | AVE. | STD. | AVE. | STD. | AVE. | STD. | AVE. | STD. | AVE. | STD. | AVE. | STD. | AVE. | STD. |
| 1-30 | Dec. | 9.7 | 1.6 | 10.0 | 1.3 | 10.0 | 1.5 | 9.8 | 1.6 | 9.8 | 1.6 | 9.8 | 1.6 | 9.8 | 1.6 | 10.1 | 1.2 | 10.1 | 1.5 | 9.8 | 1.6 | 10.2 | 1.5 | 9.9 | 1.5 | 11.3 | 1.0 | 11.5 | 1.1 |
| Winter | Jan. | 7.7 | 2.0 | 8.1 | 1.8 | 8.1 | 2.0 | 7.7 | 2.0 | 7.8 | 2.0 | 7.7 | 2.0 | 7.7 | 2.0 | 8.2 | 1.9 | 8.1 | 1.9 | 7.8 | 2.0 | 8.2 | 2.1 | 7.8 | 2.0 | 9.6 | 1.8 | 9.6 | 1.9 |
| | Feb. | 8.2 | 2.1 | 8.4 | 1.8 | 8.5 | 1.9 | 8.2 | 2.1 | 8.2 | 2.1 | 8.2 | 2.1 | 8.2 | 2.1 | 8.5 | 1.7 | 8.6 | 1.9 | 8.2 | 2.1 | 8.5 | 2.0 | 8.4 | 2.1 | 9.5 | 1.4 | 9.7 | 1.7 |
| | Total | 8.5 | 2.1 | 8.9 | 1.8 | 8.9 | 2.0 | 8.6 | 2.1 | 8.6 | 2.1 | 8.6 | 2.1 | 8.6 | 2.1 | 8.9 | 1.8 | 8.9 | 2.0 | 8.6 | 2.1 | 9.0 | 2.1 | 8.7 | 2.1 | 10.1 | 1.7 | 10.3 | 1.8 |
| | Jun. | 26.9 | 1.6 | 26.5 | 1.4 | 26.6 | 1.5 | 26.9 | 1.6 | 26.9 | 1.6 | 26.9 | 1.6 | 26.9 | 1.5 | 26.5 | 1.4 | 26.6 | 1.5 | 26.9 | 1.6 | 26.9 | 1.5 | 26.4 | 1.4 | 25.7 | 1.3 | 26.5 | 1.3 |
| Summer | Jul. | 32.0 | 3.6 | 31.3 | 3.3 | 31.5 | 3.5 | 32.0 | 3.6 | 31.9 | 3.6 | 32.0 | 3.6 | 31.9 | 3.6 | 31.3 | 3.2 | 31.5 | 3.4 | 31.9 | 3.6 | 31.9 | 3.4 | 31.5 | 3.4 | 30.2 | 2.7 | 30.9 | 3.0 |
| | Aug. | 33.2 | 2.1 | 32.8 | 1.8 | 32.7 | 1.9 | 33.2 | 2.1 | 33.2 | 2.0 | 33.2 | 2.0 | 33.2 | 2.0 | 32.9 | 1.7 | 32.8 | 1.9 | 33.2 | 2.1 | 33.2 | 2.0 | 32.5 | 2.1 | 31.6 | 1.6 | 32.5 | 1.5 |
| | Total | 30.7 | 3.8 | 30.3 | 3.6 | 30.3 | 3.6 | 30.7 | 3.7 | 30.7 | 3.7 | 30.7 | 3.7 | 30.7 | 3.7 | 30.2 | 3.5 | 30.3 | 3.6 | 30.7 | 3.7 | 30.7 | 3.7 | 30.1 | 3.6 | 29.1 | 3.2 | 30.0 | 3.3 |

**Table 5.** Average and standard deviation of the monthly indoor relative humidity in the winter and summer.

| | | Base | | Ex | | In | | 10 mm CMPCM-15 | | 20 mm CMPCM-15 | | All (10 mm CMPCM-15) | | All (20 mm CMPCM-15) | | Ex + All (20 mm CMPCM-15) | | In + All (20 mm CMPCM-15) | | Double-Layer Window | | Roof Insulation | | (ACH = 1) + NV | | Combined (Ex) | | Combined (In) | |
|---|---|---|---|---|---|---|---|---|---|---|---|---|---|---|---|---|---|---|---|---|---|---|---|---|---|---|---|---|---|
| | | AVE. | STD. | AVE. | STD. | AVE. | STD. | AVE. | STD. | AVE. | STD. | AVE. | STD. | AVE. | STD. | AVE. | STD. | AVE. | STD. | AVE. | STD. | AVE. | STD. | AVE. | STD. | AVE. | STD. | AVE. | STD. |
| 1-30 | Dec. | 68.2 | 15.2 | 66.9 | 15.6 | 66.9 | 15.1 | 68.0 | 15.2 | 67.9 | 15.3 | 68.0 | 15.3 | 68.0 | 15.3 | 66.8 | 15.7 | 66.7 | 15.2 | 68.0 | 15.3 | 66.2 | 15.4 | 67.4 | 15.4 | 61.8 | 15.9 | 61.2 | 15.5 |
| Winter | Jan. | 67.7 | 18.0 | 66.1 | 17.9 | 66.2 | 17.8 | 67.5 | 18.0 | 67.4 | 18.0 | 67.5 | 18.0 | 67.5 | 18.0 | 65.8 | 17.8 | 66.0 | 17.8 | 67.5 | 18.1 | 65.5 | 17.5 | 66.9 | 18.2 | 59.7 | 16.5 | 59.6 | 16.4 |
| | Feb. | 57.0 | 17.9 | 56.2 | 17.9 | 55.9 | 17.7 | 56.9 | 18.0 | 56.8 | 17.9 | 56.9 | 18.0 | 56.9 | 18.0 | 56.1 | 18.0 | 55.8 | 17.7 | 56.8 | 17.9 | 55.7 | 17.7 | 56.1 | 18.3 | 51.9 | 17.1 | 51.5 | 16.7 |
| | Total | 64.6 | 17.8 | 63.3 | 17.8 | 63.2 | 17.6 | 64.3 | 17.8 | 64.3 | 17.8 | 64.4 | 17.8 | 64.4 | 17.8 | 63.1 | 17.8 | 63.1 | 17.6 | 64.3 | 17.8 | 62.7 | 17.5 | 63.7 | 18.1 | 58.0 | 17.0 | 57.6 | 16.7 |
| | Jun. | 72.6 | 8.5 | 74.4 | 8.3 | 74.0 | 8.5 | 72.7 | 8.4 | 72.8 | 8.4 | 72.7 | 8.4 | 72.7 | 8.4 | 74.4 | 8.3 | 73.9 | 8.4 | 72.8 | 8.5 | 72.7 | 8.0 | 74.7 | 8.0 | 77.9 | 7.6 | 74.5 | 8.1 |
| Summer | Jul. | 65.4 | 16.1 | 67.6 | 15.3 | 67.1 | 15.7 | 65.4 | 16.0 | 65.5 | 16.0 | 65.4 | 16.0 | 65.5 | 15.9 | 67.8 | 15.0 | 67.1 | 15.6 | 65.6 | 16.1 | 65.4 | 15.4 | 67.2 | 15.6 | 71.2 | 13.7 | 68.8 | 14.4 |
| | Aug. | 52.1 | 7.6 | 53.3 | 8.0 | 53.7 | 8.0 | 52.0 | 7.5 | 52.0 | 7.5 | 52.0 | 7.5 | 51.9 | 7.6 | 53.0 | 8.0 | 53.4 | 8.0 | 52.1 | 7.6 | 51.9 | 7.5 | 54.5 | 7.6 | 56.9 | 8.3 | 54.1 | 8.3 |
| | Total | 63.3 | 14.2 | 65.0 | 14.1 | 64.8 | 14.1 | 63.3 | 14.2 | 63.3 | 14.2 | 63.3 | 14.2 | 63.3 | 14.2 | 65.0 | 14.1 | 64.7 | 14.1 | 63.4 | 14.3 | 63.2 | 13.9 | 65.4 | 13.8 | 68.6 | 13.5 | 65.7 | 13.7 |

Case 10 mm CMPCM-15, Case 20 mm CMPCM-15, Case All (10 mm CMPCM-15), Case All (20 mm CMPCM-15), Case Ex + All (20 mm CMPCM-15) and Case In + All (20 mm CMPCM-15)

As shown in Figure 11, compared to the base case, Case 10 mm CMPCM-15 and Case All (10 mm CMPCM-15) both increased $T_l$ by 0.2 °C and increased $T_r$ by up to 0.5 °C, respectively. Case 20 mm CMPCM-15 and Case All (10 mm CMPCM-15) both increased $T_l$ by 0.3 °C and increased $T_r$ by up to 0.5 °C, respectively. Besides, Case 10 mm CMPCM-15 and Case All (10 mm CMPCM-15) both reduced $RH_{in}$ by 1.7%, and Case 20 mm CMPCM-15 and Case All (20 mm CMPCM-15) both reduced $RH_{in}$ by 1.8%, respectively. In the hottest week, Case 10 mm CMPCM-15 and Case All (10 mm CMPCM-15) both reduced $T_p$ by 0.1 °C and reduced $T_r$ by up to 0.6 °C, respectively. Case 20 mm CMPCM-15 and Case All (20 mm CMPCM-15) both reduced $T_p$ by 0.1 °C and reduced $T_r$ by up to 0.6 °C and 0.7 °C, respectively. It can be seen that increasing the thickness and position of CMPCM-15 has a weak regulation effect on temperature and humidity, which is due to the fact that only 15 wt % MPCM was mixed with the material.

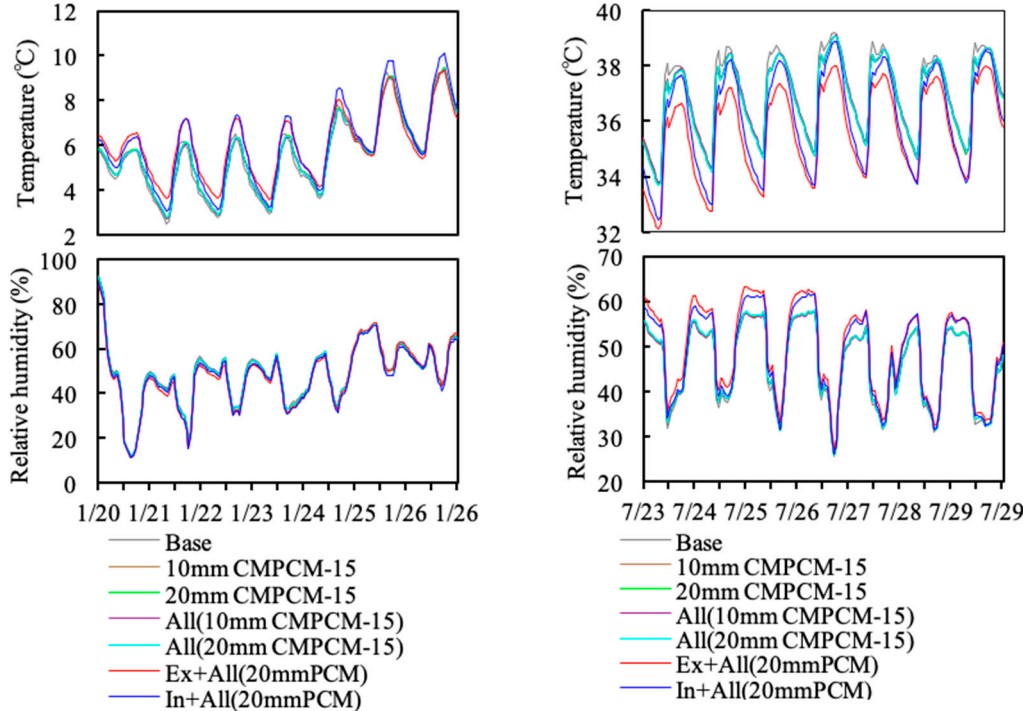

**Figure 11.** The hourly indoor temperature and relative humidity of Case 10 mm CMPCM-15, Case 10 mm CMPCM-15, Case All (10 mm CMPCM-15), Case All (20 mm CMPCM-15), Case Ex + All (20 mm CMPCM-15), and Case In + All (20 mm CMPCM-15) during the coldest week (**left**) and hottest week (**right**).

Another noteworthy phenomenon was that, when using CMPCM-15, $T_r$ increased slightly with a maximum change of 0.2 °C from night to early morning in the hottest week. This is because the melting and freezing temperature range of CMPCM-15 is [27.54 °C, 32.14 °C]. From night to early morning, the exterior walls cooled and reached the freezing temperature of CMPCM-15. CMPCM-15 was activated and started to release heat, which increased the room temperature. From morning to noon, the exterior walls heated up and reached the melting temperature of CMPCM-15. CMPCM-15 was activated and started absorbing heat, which reduced the room temperature.

In the coldest week, compared with the single external thermal insulation system, Case Ex + All (20 mm CMPCM-15) and Case In + All (20 mm CMPCM-15) further increased $T_l$ by 0.2 and 0.3 °C, respectively, and increased $T_r$ by up to 0.6 °C and 0.7 °C, respectively. Besides, Case Ex + All (20 mm CMPCM-15) and Case In + All (20 mm CMPCM-15) further reduced $RH_{in}$ by 1.7% and 2.1%. In the

hottest week, Case Ex + All (20 mm CMPCM-15) and Case In + All (20 mm CMPCM-15) further reduced $T_p$ by 0.2 °C and 0.3 °C, respectively, and reduced $T_r$ by up to 0.8 °C and 1.2 °C, respectively. Therefore, adding 20 mm CMPCM-15 layer was more effective with an internal thermal insulation system than with an external thermal insulation system.

Case double-layer window, Case roof insulation, and Case (ACH = 1) + NV

As shown in Figure 12, compared to the base case, Case double-layer window increased $T_l$ by 0.2 °C in the coldest week and reduced $T_p$ by 0.3 °C in the hottest week; this is consistent with the measurement results for the temperature. Case roof insulation increased $T_l$ by 0.9 °C and reduced $T_p$ by 0.9 °C, but only reduced $T_r$ during 61.5% of the time in the hottest week. This is because the roof receives the strongest solar radiant heat of the entire house, and this bedroom was on the top floor. Further research should be carried out to realize a balance for thermal insulation between the winter and summer in the passive design of roofs for rural residential houses in the HSCW zone. Case (ACH = 1) + NV increased $T_l$ by 0.4 °C and increased $T_r$ by up to 0.6 °C during the natural ventilation time in the coldest week. Besides, it reduced $RH_{in}$ by 4.3%. In the hottest week, Case (ACH = 1) + NV reduced $T_p$ by 0.4 °C and reduced $T_r$ by up to 1.6 °C during the natural ventilation time. In the HSCW zone, it is an effective passive means to regulate the indoor thermal environment by improving the air tightness of the house combined with natural ventilation.

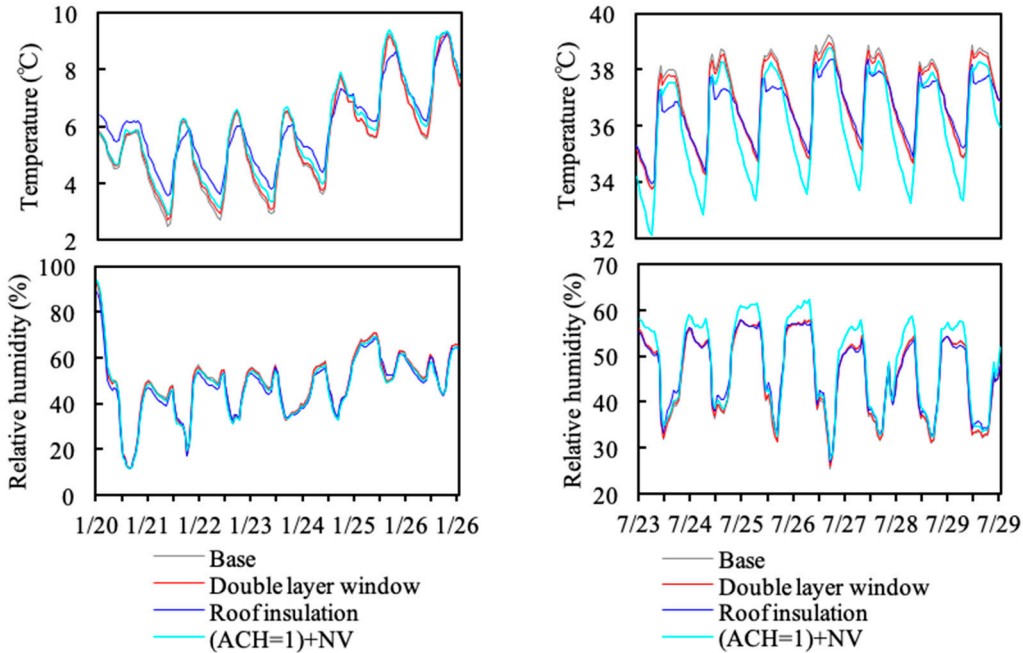

**Figure 12.** Hourly indoor temperature and relative humidity of Case double-layer window, Case roof insulation, and Case (ACH = 1) + NV during the coldest week (left) and hottest week (right).

Case Combined (Ex) and Case Combined (In)

As shown in Figure 13, Case Combined (Ex) and Case Combined (In) greatly reduced the fluctuations of the room temperature and indoor relative humidity. In the coldest week, Case Combined (Ex) and Case Combined (In) increased $T_l$ by 2.4 °C and 2.2 °C, respectively, and at most increased $T_r$ by up to 3.6 °C and 3.2 °C, respectively. Besides, Case Combined (Ex) and Case Combined (In) reduced $RH_{in}$ by 12% and 11.6%, respectively. In the hottest week, Case Combined (Ex) and Case Combined (In) reduced $T_p$ by 3.3 °C and 2.6 °C, respectively, and reduced $T_r$ by up to 4.4 °C and 3.4 °C, respectively. Regardless of the season, Case Combined (Ex) and Case Combined (In) had significantly more stable temperatures than the base case. Thus, the overall passive design of the building envelope significantly improved the indoor thermal environment.

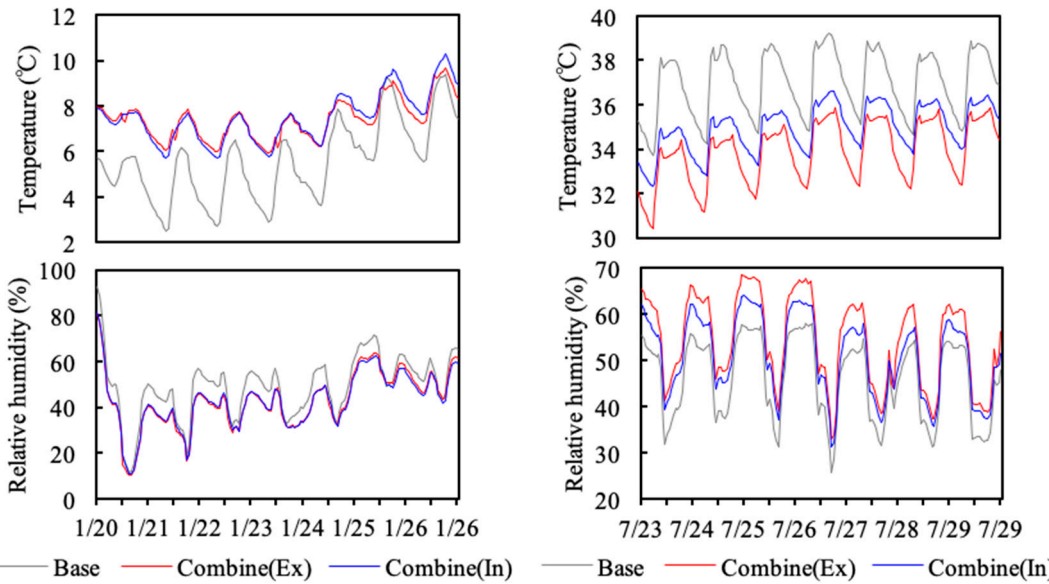

**Figure 13.** Hourly indoor temperature and relative humidity of Case Combined (Ex) and Case Combined (In) during the coldest week (left) and hottest week (right).

### 3.3.2. Energy Consumption

To compare the effects of different building envelope constructions on heating and cooling energy consumption, the energy conservation was considered in terms of the current lifestyle with the same conditions set for the equipment. The annual energy consumption of each case for the whole house was calculated. As shown in Figure 14, Case Combined (Ex) and Case Combined (In) significantly reduced the energy consumption. Case Ex, Case In, Case Ex + All (20 mm CMPCM-15), and Case In + All (20 mm CMPCM-15) also clearly affected energy consumption.

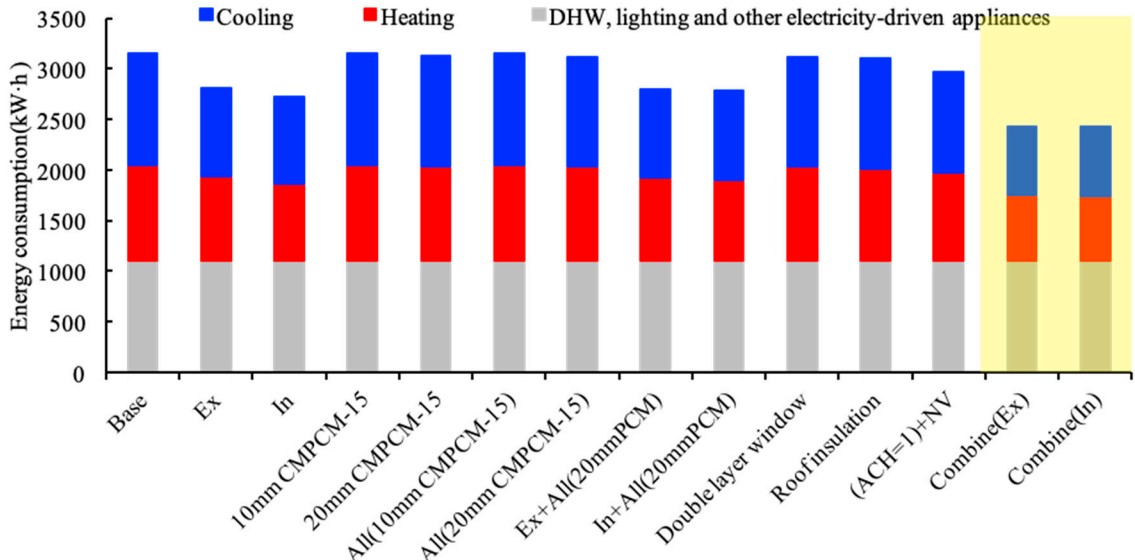

**Figure 14.** Annual energy consumption of each case for the whole house.

Table 6 presents the details of the energy consumption for each case. Case Ex and Case In reached annual energy-saving rates of 15.7% and 20.6%, respectively, which indicates their outstanding energy conservation performance. The external thermal insulation system had a superior energy conservation performance to the internal thermal insulation system in both the winter and summer. This is partly

due to the spatial and temporal energy consumption characteristics of rural residential houses in the HSCW zone. In the summer, residents typically use cooling equipment during sleeping hours (night and morning). During this period of energy use, the average temperature of the exterior wall is higher than the ambient temperature. At this time, the heat flow on the surface of the exterior wall not only dissipates heat to the outside but also dissipates heat to the room. In this case, using the external thermal insulation system hinders the heat dissipation from the wall to the outside but increases the heat dissipation from the wall to the indoor room, which increases the cooling load. With the internal thermal insulation system, the heat dissipation from the wall to the indoor room is blocked, and the heat dissipation from the wall to the outside increases, which reduces the cooling load. The same also held true in the winter.

**Table 6.** Average and standard deviation of the monthly indoor relative humidity in the winter and summer.

| No. | Case | $E_{Heating}$ [kWh] | $\Phi_{Heating}$ | $E_{Cooling}$ [kWh] | $\Phi_{Cooling}$ | $\Phi_{ENV}$ |
|---|---|---|---|---|---|---|
| 0 | Base | 944.2 | | 1110.0 | | |
| 1 | Ex | 834.6 | 11.6% | 875.0 | 21.2% | 15.7% |
| 2 | In | 763.7 | 19.1% | 859.3 | 22.6% | 20.6% |
| 3 | 10 mm CMPCM-15 | 947.4 | −0.3% | 1110.2 | 0 | −0.2% |
| 4 | 20 mm CMPCM-15Gypsum | 938.7 | 0.6% | 1094.8 | 1.4% | 0.9% |
| 5 | All (10 mm CMPCM-15) | 945.4 | −0.1% | 1105.6 | 0.4% | 0.1% |
| 6 | All (20 mm CMPCM-15) | 934.7 | 1.0% | 1086.4 | 2.1% | 1.5% |
| 7 | Ex + All (20 mm CMPCM-15) | 829.6 | 12.1% | 871.6 | 21.5% | 16.2% |
| 8 | In + All (20 mm CMPCM-15) | 797.3 | 15.6% | 885.8 | 20.2% | 17.6% |
| 9 | Double-layer window | 936.9 | 0.8% | 1077.8 | 2.9% | 1.7% |
| 10 | Roof insulation | 906.5 | 4.0% | 1097.9 | 1.1% | 2.7% |
| 11 | (ACH = 1) + NV | 878.2 | 7.0% | 993.3 | 10.5% | 8.5% |
| 12 | Combined (Ex) | 658.0 | 30.3% | 666.2 | 40% | 34.5% |
| 13 | Combined (In) | 643.7 | 31.8% | 684.1 | 38.4% | 34.6% |

The energy-saving rates of Case 10 mm CMPCM-15, Case 20 mm CMPCM-15, Case All (10 mm CMPCM-15), and Case All (20 mm CMPCM-15) were −0.2%, 0.9%, 0.1%, and 1.5%, respectively. It can be seen that increasing the thickness and position of CMPCM-15 has a very weak effect on energy conservation, which is also due to the fact that only 15 wt % MPCM was mixed with the material. In general, the internal thermal insulation system (CASE In + All (20 mm CMPCM-15)) saved more energy than the external thermal insulation system (Case Ex + All (20 mm CMPCM-15)), with an energy-saving rate of 16.2% versus 17.6%. This was also due to the spatial and temporal energy consumption characteristics of rural residential houses in the HSCW zone.

The annual energy-saving rate of Case double-layer window (1.7%) indicated that double-glazed windows provided superior thermal insulation to that of single-glazed windows. The annual energy-saving rate of Case roof insulation showed good energy conservation (4.0%) in the winter. Case (ACH = 1) + NV effectively reduced the energy consumption because the air infiltration between indoors and outdoors was reduced, which had an energy-saving rate of 8.5%.

The annual energy-saving rates of Case Combined (Ex) and Case Combined (In) were 34.5% and 34.6%, respectively. Thus, when the indoor temperature is kept the same with air conditioning, an overall passive design can greatly improve the energy conservation of rural residential houses in the HSCW zone.

## 4. Conclusions

Face-to-face questionnaires and interviews, temperature and humidity monitoring, and air-tightness testing were carried out to evaluate rural residential houses in the HSCW zone. The following was concluded:

- Rural residential houses in the HSCW zone have a very poor thermal insulation performance. The exterior walls were mostly made of 240 mm thick brick, and up to 90% of exterior walls have no insulation. In addition, 96% of the houses used single-glazed 3 mm thick glass windows. A pitched roof with a wooden structure and no insulation is used.
- Rural residential houses in the HSCW zone have high energy consumption costs. The average household electricity fee reaches 139.2 CNY/month, and the average gas fee per household is 75 CNY/month.

  Passive design measures for rural residential houses in the HSCW zone indicated the following:

- Increasing the thickness and position of CMPCM-15 has a regulation effect on the indoor thermal environment and saves energy. Adding the 20 mm CMPCM-15 layer to both the exterior wall and interior wall increased the room temperature by up to 0.5 °C in winter and reduced the room temperature by up to 0.7 °C in summer, and it can reduce indoor relative humidity by 1.8% in winter. The energy-saving rate of adding a 20 mm CMPCM-15 layer to both the exterior wall and interior wall reached 1.5%.
- The overall passive design greatly reduced the fluctuations of the room temperature and indoor relative humidity. In the winter, the overall passive design combined with the external/internal insulation system of the exterior wall increased the room temperature by up to 3.6 °C/3.2 °C, respectively, and reduced the indoor relative humidity by 12%/11.6%, respectively. In the summer, the overall passive design combined with the external/ internal insulation system of the exterior wall reduced the room temperature by up to 4.4 °C/3.4 °C, respectively. The annual energy-saving rates of the overall passive design combined with the external/ internal insulation system of the exterior wall were 34.5%/34.6%, respectively.

**Author Contributions:** Conceptualization, H.Z. and J.R.; methodology, H.Z. and J.R.; software, J.R.; validation, H.Z. and J.R.; formal analysis, J.R.; investigation, H.Z., J.R., C.S., Y.C.; resources, H.Z. and C.D.; data curation, H.Z. and J.R.; writing—original draft preparation, J.R.; writing—review and editing, H.Z., J.R. and D.P.; visualization, H.Z.; supervision, H.Z.; project administration, H.Z.; funding acquisition, H.Z.

**Funding:** This research was funded by the National Natural Science Foundation of China (51778358).

**Acknowledgments:** The authors thank the staff of the Anji County Planning Bureau and Dazhuyuan Village Committee, as well as the occupants of the investigated dwellings for their helpful cooperation.

**Conflicts of Interest:** The authors declare no conflict of interest.

## Abbreviations

The following abbreviations are used in this manuscript.

| | |
|---|---|
| HSCW | hot summer and cold winter zone |
| $U$ | moisture content defined as the mass fraction of water contained in a material |
| $\varphi$ | surface air relative humidity |
| $a, b, c, d$ | coefficients that define the relationship between a material's moisture content and the surface air relative humidity |
| $E_{bld, des}$ | comprehensive energy consumption for annual heating and cooling of the design building |
| $E_{bld,ref}$ | comprehensive energy consumption for annual heating and cooling of the reference building |
| $T_r$ | room temperature |
| $RH_{in}$ | indoor relative humidity |
| $T_l$ | lowest room temperature during the coldest week |
| $T_p$ | peak room temperature during the hottest week |

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
