# Peer review of "Survey on the Indoor Thermal Environment and Passive Design of Rural Residential Houses in the HSCW Zone of China"

_sustainability, doi:10.3390/su11226471_

Round 1

Reviewer 1 Report

The present paper is well structured and rich in information: both questionnaires and simulations of IEQ parameters are carried out to inform the design options of rural houses in the HSCW zone in China.

However, in order to make the paper publishable, the following major amendments are needed in the reviewer's opinion:

there are too many figures (nineteen) that make the readability poor and distract the reader from the main findings. I suggest cutting down some of them (from Fig. 10 onwards)  introduction: an aspect that I saw disregarded is the role of natural ventilation in this climate zone, which is pivotal in providing comfort conditions. I suggest referring to "V. Costanzo, R. Yao, T. Xu, J. Xiong, Q. Zhang, B. Li, Natural ventilation potential for residential buildings in a densely built-up and highly polluted environment. A case study, Renewable Energy 138 (2019), 340-353" where the NV potential is worked out for both polluted and not-polluted environments Fig.2 is not clear, please amend or consider removing it Table 1: it is not clear where the material properties are gathered from. Please clearly reference their source (eg. technical standards) please check and amend all the references in Chinese language in the text it is not 100% clear if DHW and other electricity-driven appliances consumption figures are embedded in both bills and simulations. Please explain it Table 3 lists all the simulated scenarios, but I do think some of them are not really necessary (as demonstrated also by the results). In this light, I suggest getting rid of cases 3, 5 and 7. This will also make all the graphs more understandable Table 5: I suggest to highlight the best/worst cases, and do the same also in Fig. 19 conclusions should also better highlight the best configuration found via simulations getting rid of acronyms and using full names and explanation of the specific case as a side note, although not fundamental for the improvement of the paper, I will encourage the authors to run more specific analysis concerning the use of PCM following what already done in "V. Costanzo, G. Evola, L. Marletta, F.Nocera, The effectiveness of phase change materials in relation to summer thermal comfort in air-conditioned office buildings, Building Simulation 11 (6) (2018), 1145-1161", where several combinations of thickness/position/melting temperatures are explored. Alternatively, I suggest mentioning that the cases using PCM are not optimal since their performance is affected by the all the previous factors

Reviewer 2 Report

The manuscript aims to understand the current status of the indoor thermal environment for rural residential houses in the HSCW zone and analyze its cause in order to develop some strategies for improvement through passive design of the building envelope.

The methodology used in the study includes questionnaire survey, field measurement and simulations. The survey methodology was not described in detail. From the survey results is shown in Section 3.1, it seems the survey did not include questions on the occupant’s perception in relation to thermal comfort, which is very important for this kind of study. The subjective measurement is missing in this study. As the results, the field measurement data was analyzed according to ASHRAE 55 only, without comparing it with occupant’s perception.

In the manuscript, it is also not clear on how the field measurement data was used to support the simulation. The data can be used as validation of the existing building design in the simulation, before conducting different simulation scenarios.

Round 2

Reviewer 1 Report

The authors have done an extensive revision from the original manuscripts and I have no further comments

Reviewer 2 Report

The authors successfully addressed the reviewer's concerns risen during the first submission by significantly improving the quality of the manuscript. I think the paper does not need any further revision and is thus ready for publication